# Conversations about FGM in primary care: a realist review on how, why and under what circumstances FGM is discussed in general practice consultations

Sharon Dixon ![ORCID],[1] Claire Duddy ![ORCID],[1] Gabrielle Harrison,[2] Chrysanthi Papoutsi,[1] Sue Ziebland,[1] Frances Griffiths ![ORCID] [3,4]

► Prepublication history and supplemental material for this paper is available online. To view these files, please visit the journal online (http://dx.doi.org/10.1136/bmjopen-2020-039809).

[1]Nuffield Department of Primary Care Health Sciences, University of Oxford, Oxford, UK
[2]University Hospitals Coventry and Warwickshire, NHS Trust, Coventry, UK
[3]Centre for Health Policy, University of the Witwatersrand, Johannesburg, South Africa
[4]Warwick Medical School, University of Warwick, Coventry, UK

**Correspondence to**
Dr Sharon Dixon;
sharon.dixon@phc.ox.ac.uk

## ABSTRACT

**Objectives** Little is known about the management of female genital mutilation (FGM) in primary care. There have been significant recent statutory changes relevant to general practitioners (GPs) in England, including a mandatory reporting duty. We undertook a realist synthesis to explore what influences how and when GPs discuss FGM with their patients.

**Setting** Primary care in England.

**Data sources** Realist literature synthesis searching 10 databases with terms: GPs, primary care, obstetrics, gynaecology, midwifery and FGM (UK and worldwide). Citation chasing was used, and relevant grey literature was included, including searching FGM advocacy organisation websites for relevant data. Other potentially relevant literature fields were searched for evidence to inform programme theory development. We included all study designs and papers that presented evidence about factors potentially relevant to considering how, why and in what circumstances GPs feel able to discuss FGM with their patients.

**Primary outcome measure** This realist review developed programme theory, tested against existing evidence, on what influences GPs actions and reactions to FGM in primary care consultations and where, when and why these influences are activated.

**Results** 124 documents were included in the synthesis. Our analysis found that GPs need knowledge and training to help them support their patients with FGM, including who may be affected, what needs they may have and how to talk sensitively about FGM. Access to specialist services and guidance may help them with this role. Reporting requirements may complicate these conversations.

**Conclusions** There is a pressing need to develop (and evaluate) training to help GPs meet FGM-affected communities' health needs and to promote the accessibility of primary care. Education and resources should be developed in partnership with community members. The impact of the mandatory reporting requirement and the Enhanced Dataset on healthcare interactions in primary care warrants evaluation.

**PROSPERO registration number** CRD42018091996.

## Strengths and limitations of this study

► A realist approach to synthesis facilitated inclusion of a wide range of data sources and consideration of this research question despite little direct primary care research about female genital mutilation, with a comprehensive and iterative approach to data searching for relevant evidence.

► This method facilitated the inclusion of community, charity and advocacy organisation data contributing evidence that might not have been accessible using other methods.

► We searched widely for data to inform the question in comparative fields.

► The tabulation and characterisation of the published research are themselves valuable and highlight potential research gaps.

## INTRODUCTION

Female genital mutilation (FGM) is defined as all procedures that intentionally alter or cause injury to the female genitalia for non-medical reasons. There are no known health benefits and many documented harms, including immediate and long-term physical and psychological consequences. FGM is recognised internationally as an act of violence against women and girls. FGM is categorised into four types: type I (clitoridectomy), type II (partial or total removal of the clitoris and labia minora/majora), type III (infibulation) and type IV (all other harmful procedures, including pricking and piercing).[1]

UNICEF estimates that 200 million girls and women in 30 countries worldwide have been subjected to FGM. Global migration from areas where FGM is traditionally practised means that FGM is now a worldwide health concern.[2]

In 2011, it was estimated that 137 000 women and girls with FGM from countries

where FGM is traditionally practised were permanently resident in England and Wales. Prevalence of FGM was thought to be highest in urban areas, but women and girls affected by FGM were likely to live in every local authority area in England and Wales.[3]

In 2014, the UK government hosted the first Girl Summit, in partnership with UNICEF, at which they pledged to mobilise domestic and international efforts to stamp out FGM within a generation, launching a raft of initiatives, including a £1.4 million FGM prevention programme with NHS England, and legislative changes.[4] The FGM prevention programme sought to improve how the NHS responds to FGM,[5] and sets out expectations for NHS staff, including about data recording.[6]

FGM has been a specific offence in the UK since 1985.[7] Before 2019 there had been no UK convictions, with the lack of progress tackling FGM described as a 'national scandal' in 2014 by a Home Affairs Select Committee reporting on the case for an FGM National Action Plan (with aims including achieving a successful prosecution and improving safeguarding and services).[8] In 2015, a mandatory reporting duty was introduced in England and Wales, requiring all registered professionals to report all cases in under 18 year olds where FGM was identified on examination or through a first-hand disclosure directly to the police.[9] Additionally, an FGM Enhanced Dataset was introduced in 2015 in England, mandating the submission of quarterly data returns, including personally identifiable data from all general practitioner (GP) practices.[10] Data return rates from primary care to the Enhanced Dataset have been low, with only 64 GP practices in England submitting data returns in 2018–2019.[11] The reasons for this are not known. Concerns have been raised by clinicians and community members about the potential impacts of mandatory reporting and the Enhanced Dataset on trust and patient–doctor relationships.[12–15]

In this realist synthesis, we seek to understand factors that can potentially influence how GPs and women and girls from FGM-affected communities interact in English primary care in the current UK context. This was identified as an important area for exploration in a research user consultation where community members and professionals were asked what they identified as FGM research and service priorities.[16] General practice care in England is typically delivered by primary care health teams, including GPs, practice nurses, advanced nurse practitioners (ANPs) and, increasingly, pharmacists and paramedics who are based in GP practices located within community settings. Primary care in England holds patients and families in holistic and longitudinal care (and care records) and has a gatekeeper role. Prior systematic reviews have shown that around the world, health professionals do not have adequate knowledge about FGM, although these reviews primarily focused on obstetrics and gynaecology.[17 18]

Our exploratory review identified the relevant literature as disparate and heterogeneous. Therefore, we identified a need for bringing together and making sense of different types of evidence that would help develop our understanding of how, why and under what circumstances FGM is discussed (or not) in primary care consultations in England, in the context of recent policy changes.

To explore this overarching review aim, we identified the following review questions:

1. What influences how GPs manage FGM in their clinical practice and why?
2. What influences GPs' actions when they consider initiating discussion about FGM with patients in primary care? Where, when and why are these influences active?
3. What influences how GPs respond to a patient-initiated disclosure of FGM during a primary care consultation? Where, when and why are these influences active?

## METHODS

For GPs, supporting women with FGM and managing the attendant reporting, safeguarding and clinical needs associated with this can be viewed as a complex intervention (defined by the Medical Research Council as an intervention with several interacting components, and where the behaviours required by the intervention are numerous or complex[19]).The FGM Prevention Programme included the provision of new educational materials, safeguarding resources and new obligations to document and report FGM. This requires GPs to participate in educational opportunities, consider when and how to discuss FGM with their patients, and consider when and how they need to comply with reporting requirements.

Realist synthesis is a theory-driven and interpretive systematic review methodology with an explanatory rather than judgemental focus which can be used to evaluate the impact of complex policy. Adopting a realist synthesis methodological approach, the research question does not only explore whether an intervention works—or not—but explicitly considers under what circumstances (when, why and how) an intervention might generate outcomes. A realist synthesis seeks to explore the contexts under which outcomes occur, and the mechanisms (processes which connect the context and the outcome) which link them.[20]

There is little primary research about how English GPs are managing their patients with FGM. The realist review approach as defined by Realist and Meta-narrative Evidence Syntheses Evolving Standards (RAMESES) allows development of programme theory based on evidence about managing FGM in other healthcare settings (eg, obstetrics and midwifery), from grey literature, including opinion pieces and charity publications, and for testing evolving programme theory in potentially comparable healthcare challenges in English primary care.

We used the RAMESES publication standards to develop[21] and report this realist synthesis,[22] and followed methodology described in other realist syntheses.[23] The

study was registered on prospero and the protocol is available in online supplemental appendix 1.

## Patient and public involvement (PPI)

This synthesis was developed following a patient and public involvement research priority setting project, in which SD was involved in developing and reporting, which identified this question as a research priority.[16] The findings of this synthesis were reviewed with stakeholders and PPI collaborators who commented on resonance and relevance.

## Initial programme theory development

An initial programme theory (a theory describing how and why the interventions being considered are hypothesised to operate to generate outcomes)[22] considering how and under what circumstances GPs in England might talk about FGM with their patients was derived by SD based on an exploratory literature review and relevant policy documentation (online supplemental appendix 2). SD works as a GP partner and is a practice safeguarding lead. Since 2015, SD has been a primary care member of her local safeguarding board FGM operational group. SD is a trustee of Oxford Against Cutting. This synthesis was developed by SD following a public and stakeholder research priority setting project which identified service priorities (including the need for holistic services throughout the life course) and research questions, including the potential impacts of mandatory reporting and the FGM Enhanced Dataset on trust in healthcare. Following the commencement of this literature synthesis, SD developed and began a qualitative study undertaking interviews with GPs about their perspectives on supporting women affected by FGM in the context of English primary care following the introduction of mandatory reporting and the FGM Enhanced Dataset. An expert advisory group consisting of six primary healthcare professionals (HCPs), including local and national FGM experts, acted as project stakeholders and advised on how the programme theory fitted within primary care processes, a recognised contribution to realist synthesis.[24]

## Searching

Searching for data for inclusion in this review was conducted in multiple stages. Initial exploratory scoping searches identified an initial set of relevant documents which informed the development of the initial programme theory and contributed to theory refinement. The main searches to identify evidence to test our programme theory were developed by an information specialist and conducted in the following databases: MEDLINE, Embase, PsycINFO, Global Health, CINAHL, Web of Science, Sociological Abstracts, Anthropology Plus, Social Science Abstracts and ASSIA. We searched for literature on health professionals and FGM in the UK and worldwide, in primary care and obstetrics and gynaecology, and we searched for international qualitative studies and reviews on FGM. There was no time limit set for the search. We included other health settings where the reporting and communication requirements might be similar because of the lack of primary care data. We built our dataset iteratively by searching reference lists and conducting forward citation searches for key papers. We included grey literature identified in this way and used recommendations from experts in FGM, including systematically searching for reports from English FGM advocacy organisations. We conducted update searches to ensure that the most recent evidence was included. We also conducted secondary searches in the following bodies of literature, identified as areas where comparable contexts and mechanisms may occur to inform our developing programme theory: intimate partner violence (IPV) in primary care, Driving & Vehicle Licensing Agency (DVLA) reporting, the Prevent programme and mandatory reporting (in health and education). The searches are available in the online supplemental file (online supplemental appendix 3).

## Selection and appraisal of documents

We included all study designs and papers that presented evidence about factors potentially relevant to considering how, why and in what circumstances GPs feel able to discuss (or not) FGM with their patients in English primary care settings, following the recent regulatory changes. Where documents from non-UK settings were identified, we excluded those written in languages other than English and from contexts likely to be significantly different from UK primary care. We included opinion pieces in influential UK medical journals as these could provide explanations relevant to UK primary care. All titles and abstracts were screened by SD and were included for full-text review if they were judged as potentially containing information relevant to conversations about FGM in primary care. Papers that contained information possibly informative about any part of the process of primary care consultation were considered. Our scoping had shown us that there was no primary evidence that covered all aspects of our review question, and so we identified that to having broad and open criterion would facilitate the development of responsive and wide-ranging programme theory. In keeping with realist methodology, data selection and inclusion were iterative, beginning with our initial programme theory and then revisited as new areas of theory emerged. For example, as emotive responses to FGM emerged as an important relevant factor from our initial searching, we revisited our search for evidence to explore this, as is established in realist synthesis methodology.[20] In keeping with realist synthesis methodology, papers were appraised for their relevance (whether the study provided evidence relevant to the theory under development) and rigour (whether the inferences within the evidence make a credible contribution to testing a developing theory). The concept of data saturation is applicable within realist synthesis.[20] Together, these methodological parameters made up the study framework, which determined inclusion and exclusion. Ten per

cent of selected abstracts were independently reviewed by GH/CD and discrepancies were discussed to inform the remaining screening process. All included full-text documents were reviewed and inclusion was agreed with GH/CD/study team.

## Data extraction and analysis

Papers selected for full text review were loaded into NVivo and coded. We based initial codes on the initial programme theory, adding further emergent codes during analysis. We examined the coded data for patterns (demiregularities[25]) from which we identified potential contexts, mechanisms and outcomes of interest. These were iteratively examined within the evidence to develop configurations of contexts, mechanisms and outcomes (CMOCs). The expert advisory group commented on the resonance of the developing and finally presented CMOCs. We prioritised high-quality empirical data; however, we included evidence from other sources if it was relevant to the developing primary care CMOCs.[20 26] The resulting programme theory was developed in realist format depicting the relationship between context, outcome and the mechanism linking them. Developing theory was tested for resonance by considering it against evidence from comparative literature fields.[24]

## RESULTS

In the searching directly related to FGM (including the scoping, main and update searches), 4035 abstracts were identified and screened, leading to a full-text review of 346 papers, with an additional 51 papers added for full-text review from recommendation, advocacy organisation searching and citation tracking, leading to the final inclusion of 94 papers from the FGM literature. Comparative literature was identified from the secondary searches from UK primary care in IPV, the Prevent strategy, DVLA reporting and mandated reporting. Reading within the Prevent literature, we identified evidence related to data sharing in the context of immigration enforcement, and this was also included where relevant; these searches identified 593 citations, of which 121 were included for full-text review, and 30 informed the synthesis. Figure 1 summarises the search strategy; table 1 summarises the characteristics of the included FGM evidence and table 2 summarises the characteristics of the included evidence from the comparative literature.

This section is a narrative account of the final programme theory (explanatory configurations of contexts, mechanisms and outcomes) developed by this synthesis considering how, why and under what circumstances GPs might initiate (or not) or respond to FGM (or not) in the setting of English primary care. A full set of CMOCs with supporting data is available as a supplementary file (online supplemental appendix 4). The programme theory is presented within four themes: FGM knowledge and awareness, communication about FGM, the role of guidelines and service provision, and mandatory reporting requirements. The overarching realist

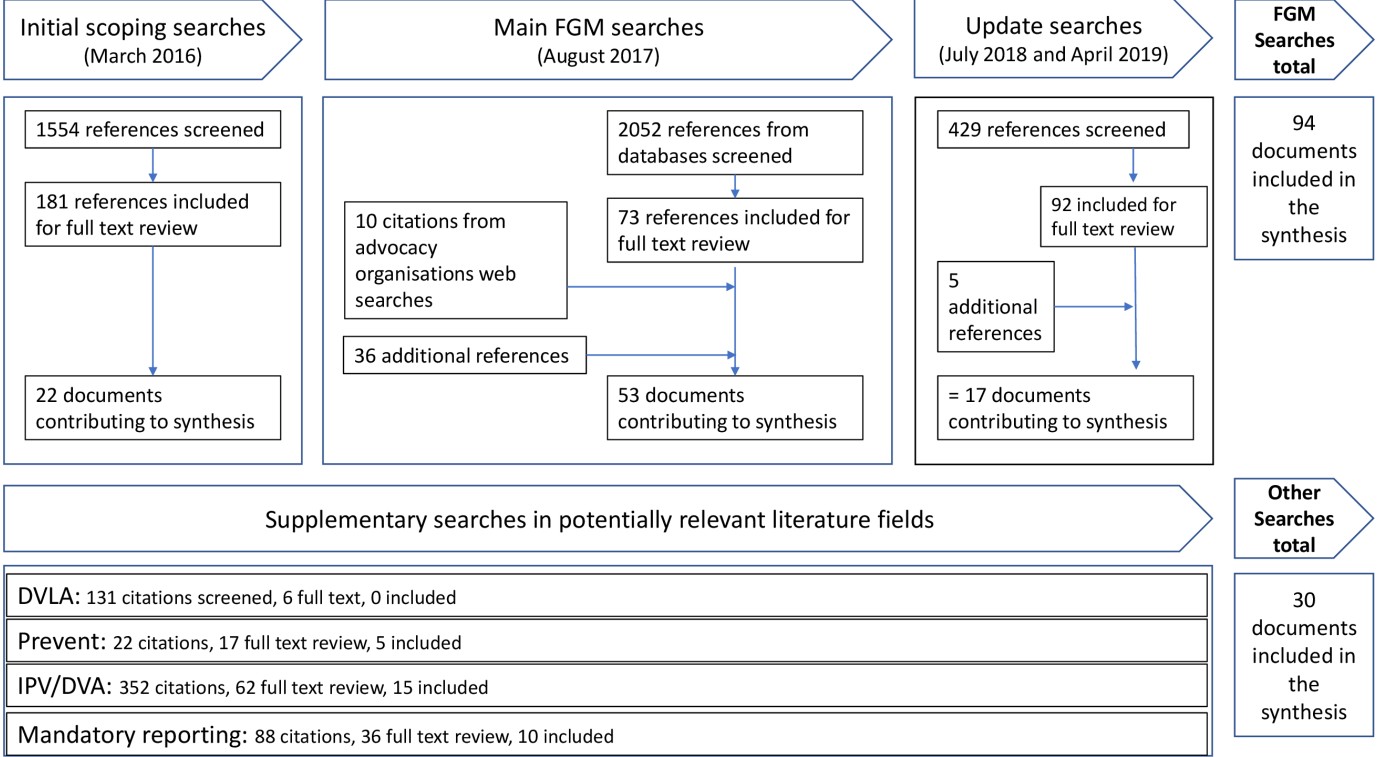

**Figure 1** Summary of our searching and screening processes. Tables 1 and 2 summarise the characteristics of the papers included in the synthesis. DVA, domestic violence and abuse; DVLA, Driving & Vehicle Licensing Agency; FGM, female genital mutilation; IPV, intimate partner violence.

**Table 1** Characteristics of the 94 papers making up the FGM literature evidence which contributed to CMOC derivation

| Type of literature | Papers (n) |
|---|---|
| Systematic/literature review | 12 |
| Qualitative | 32 |
| Quantitative/survey/questionnaires | 15 |
| Audit/case series | 9 |
| Charity reports | 9 |
| Other (eg, editorial opinion piece) | 17 |
| Country of research origin | |
| UK | 35 |
| Other European | 21 |
| America/Canada | 5 |
| Australia | 4 |
| Africa | 1 |
| Whose perspective | |
| Provider | 29 |
| Community | 28 |
| Both | 10 |
| Details of providers | |
| Obstetrics/gynaecology / midwifery | 21 |
| Other secondary care | 11 |
| Mixed, including primary care | 6 |
| Country of origin of women research conducted with/included | |
| Somalia/infibulated women/type III FGM | 23 |
| Mixed African | 12 |
| Mixed | 2 |

CMOC, context, mechanism and outcome; FGM, female genital mutilation.

**Table 2** Characteristics of the 30 papers included from literature identifies in secondary searches

| Characteristics | Papers (n) |
|---|---|
| Intimate partner violence | |
| Primary qualitative | 8 |
| Primary quantitative | 2 |
| Other/opinion/guidance/review/ service evaluation | 5 |
| Mandatory reporting | |
| Primary qualitative | 4 |
| Primary quantitative | 4 |
| Other/opinion/guidance/review | 2 |
| Prevent strategy/data sharing | |
| Primary qualitative | 2 |
| Primary quantitative | |
| Other/opinion/guidance/review | 3 |

programme theory is illustrated graphically at the conclusion of the results section.

### FGM knowledge and awareness

Health professionals need to have adequate knowledge about FGM to meet the care needs of patients who may be affected by FGM (including having had FGM or potentially being at risk of FGM).[17 18 27–47] Awareness of who might be affected by FGM, as well as the different types of FGM and their associated clinical consequences, will influence whether GPs identify a need to consider or ask about FGM with their patients. Health professionals also need to know about relevant legislation, and their statutory and safeguarding requirements in relation to FGM.[17 18 33 34 38 41 47–49]

Whether GPs are aware (or not) of what they may not know about FGM, lacking knowledge about FGM affects practitioners' ability and confidence when caring for women with FGM,[36 50 51] including confidence to consider who may be at risk.[52] Knowing how to respond to a disclosure or when identifying that a woman has FGM may help GPs feel confident to ask.[53 54] In turn, women who perceive that GPs do not have the knowledge or skills to recognise their FGM-related care needs, or who feel potentially stigmatised because of their FGM, may lack confidence in accessing healthcare.[31 36 50 55–59]

Health professionals report experiencing strong emotional reactions to encountering FGM, including anger, shock and pity,[47 60–62] and that seeing FGM without having adequate knowledge can be 'frightening'.[61] Experiencing these strong emotional responses may contribute to clinicians feeling panicked, and abandoning usual practices and routines.[54 63] Although professionals describe trying to hide their reactions, they were aware that they may be apparent to the women.[45 61] This observation is mirrored by evidence from community members who describe feeling ashamed or judged when HCPs react with shock or horror to their FGM, notably during physical examinations.[55 64–68] This can impact on their willingness to access services,[46 64–67] including attending for cervical smears.[69] This could be mitigated against when professionals were able to act with confidence and sensitivity.[54 67 70]

Health professionals with experience of supporting patients with FGM were likely to have more knowledge about FGM.[34 38 40 71] A potential challenge for GPs is that FGM may form only a small part of their workload,[14 72 73] meaning that learning about FGM may not be identified as a priority.

Another potential difficulty is that FGM can be challenging to identify.[74] Recent data from a specialist paediatric clinic in England noted that the examination signs in type IV can be subtle and potentially difficult to identify,[75 76] or be associated with minimal or non-specific symptoms.[75 77] Others have noted that not all GPs will have the necessary expertise to identify all types of FGM.[12 78] Added to this complexity is that self-reporting by women about their FGM type in health settings has been shown

not to be reliable,[79] which could impact on how women feel able to respond to some questions about FGM.

## Talking about FGM and communication

A key skill GPs need is being able to talk about FGM sensitively.[36 45 80] GPs who do not feel confident in raising the subject of FGM, for example, because they are worried that they will upset or offend women, may avoid talking about FGM.[14 30 42 46 47 49 55 61 64 74 80 81] Health professionals who perceive that discussing FGM can be culturally taboo or a sensitive subject could also be fearful of offending women reported sometimes avoiding talking about FGM,[14 47 80 82] as may women.[55 83] This was described in one study as an 'expression of respect'.[80] This contextual factor may be evolving as community attitudes towards talking about FGM are shifting,[48 63 84 85] including that talking about FGM is becoming less taboo in some communities.[86–88] Lacking awareness of shifting community attitudes can risk offending or alienating members of those communities,[28 48 63 85] which risks reducing effective communication.

The words or terminology that GPs use when they talk about FGM can complicate communication with their patients. For example, if the term FGM is not familiar to the woman,[89 90] offends or alarms her,[50 91] or she does not align her cultural practice (eg, labial elongation) with FGM,[92] then she may not relate her experience to FGM. This can complicate conversations about the potential health consequences of FGM or whether women perceive the conversation is relevant to their experience of FGM.[93]

Women who were aware of previous difficult experiences with communication and engagement with health professionals (eg, language, cultural differences, perceived judgement) where they had not felt understood or respected describe a lack of confidence and trust in health services.[45 55–57 64 94] Women who feel pitied or judged may be less likely to feel able to make a disclosure to HCPs.[95 96] Language barriers or a lack of understanding about how health services work, including negotiating with primary care reception staff can complicate access to healthcare.[97]

A potential strategy that could help facilitate both the acceptability and accessibility of services is the involvement of community health advocates, such as members of FGM-affected communities, who can act as a bridge between communities and services, for example, by promoting trust and providing education for both community members and health professionals.[16 98 99]

For GPs, whether FGM is raised by the patient or the GP, an important contextual factor is whether FGM is perceived to be relevant to the health concern which the woman brings to her GP appointment. This can influence both whether the subject of FGM is broached, and then if broached, how it is received and experienced.[48 62 84 90 100] Women who feel that the HCP is preoccupied with their FGM, rather than their health concerns, may disengage from the healthcare setting.[48 84 101 102] This is also potentially relevant when GPs consider asking women about

their experience of FGM with the aim of considering the safeguarding needs within their families, rather than the woman's own health needs.[48 53 63 73 103 104] Balancing the needs of the woman who is presenting with the potential needs of her wider family may introduce complex considerations for GPs when they are considering how and into whose medical notes they code FGM into primary care medical records.[104–106]

An important context which influences how able (and enabled) GPs and their patients are to effectively communicate about FGM is whether there is a language barrier between them or not.[14 18 30 45 47 55 81 82 97 107–109] Strategies to address language barriers add their own complications. Official interpreters are recommended, but may not be available or trusted by women, for example, if they both perceive FGM as taboo, or she fears they will not respect her confidentiality. This can lead to fear and reduced engagement with health professionals.[14 30 32 47 49 61 62 69 110 111] The presence of family members (as interpreters or witnesses) in the consultation may inhibit GPs from feeling able to raise FGM with the women because of concerns about privacy and confidentiality.[14 30 82 112]

Finally, factors such as the GP's gender may influence whether the woman or GP feel it is culturally appropriate to talk about FGM.[14 47 69 107] Time constraints in the consultation may act as a potential barrier which deters GPs when they are contemplating discussing FGM.[14 47 50 73 113]

## Need for guidelines and access to specialist services

Researchers and commentators suggest that having access to clear guidelines will enable professionals to ask women about FGM and optimise their care.[17 44 47 61 63 114] However, even when guidelines exist, awareness of them may be incomplete or they may not be followed, as demonstrated by four UK hospital studies.[49 71 81 115] Access to guidance and specialist services may be especially important for clinicians who see FGM less often.[47] Lacking guidance, including a lack of certainty of what 'good care' comprises, can lead clinicians to feel unsure and to improvise how they offer care,[46 47] and risks ethical dilemmas or incorrect decision making.[63 114]

Normalising asking about FGM, for example, by using prompts in the medical record, may overcome some clinician barriers and help them begin these discussions,[116] especially if these are then linked to resources or care pathways.[117]

Having access to services which could offer specialist assessments, treatment or advice may help GPs feel enabled to raise the subject of FGM.[12 47 55 57 78 118] When health professionals speak about FGM within a framework of offering support and services, it is more likely to be experienced as acceptable by their patients.[88] When training and education are supported by referral pathways or protocols for intervention, they are more likely to be effective in changing behaviour and promoting clinicians asking.[54 95 117 119]

## Mandated actions including the mandatory reporting duty and the FGM Enhanced Dataset requirements

It is unknown what impact the FGM reporting duties have on healthcare interactions, but concerns have been raised that approaches to safeguarding including the mandatory reporting duty in FGM may cause women distress or reduce their trust in HCPs, which may deter women from seeking healthcare or disclosing their FGM to HCPs.[12 13 84 86 120 121] Furthermore, if women perceive that the HCP is more interested in data management about their FGM than their needs, they may feel disrespected and may avoid healthcare settings.[84]

The requirement to send personally identifiable data to the FGM Enhanced Dataset was also identified as a potential barrier to talking about FGM by both women and GPs.[14–16 84 122 123] When women are not confident that their medical encounters or records are confidential, they may feel fearful and avoid making disclosures.[112 124]

The concern that making a mandated report would have a negative impact on ongoing effective doctor–patient relationships may be an important consideration for professionals,[125–129] including whether this might deter patients from accessing services.[126 130] Lacking confidence that making a mandated report will be met with an acceptable or adequate response may pose challenges for HCPs.[128 131]

Practitioners may need to feel certain before making a mandated report,[128 132 133] and challenges in identifying less apparent forms of FGM may add tensions to the requirements for mandated reporting.[128] In the case of FGM, this could be complicated if GPs do not feel confident that they have the knowledge or skills to correctly identify FGM,[74 75] especially types I and IV, which may be both harder to visualise on examination and more commonly encountered.[75 76 134 135] Mandatory reporting by health professionals should be supported by educational resources and training;[133 135] reporting without training may lead to inaccurate data collection.[136] Clinician concerns about confidentiality and stigma could also contribute to incomplete or inaccurate data coding.[137]

When young people know that the professional whom they are speaking to is mandated to share the information with other authorities, they may feel more reluctant to trust the professional and may be less likely to make a disclosure.[138] Those who are potentially fearful of authorities or perceive themselves to be vulnerable, for example, those with uncertain migrant status, may be more fearful of mandatory reporting or data sharing and may avoid accessing services.[139–143]

### Overarching programme theory

Figures 2 and 3 bring together the emerging CMOC configurations that made up the programme theory for this review and depict a theoretical framework for how these might inter-relate, thus offering a conceptual summary of this synthesis.

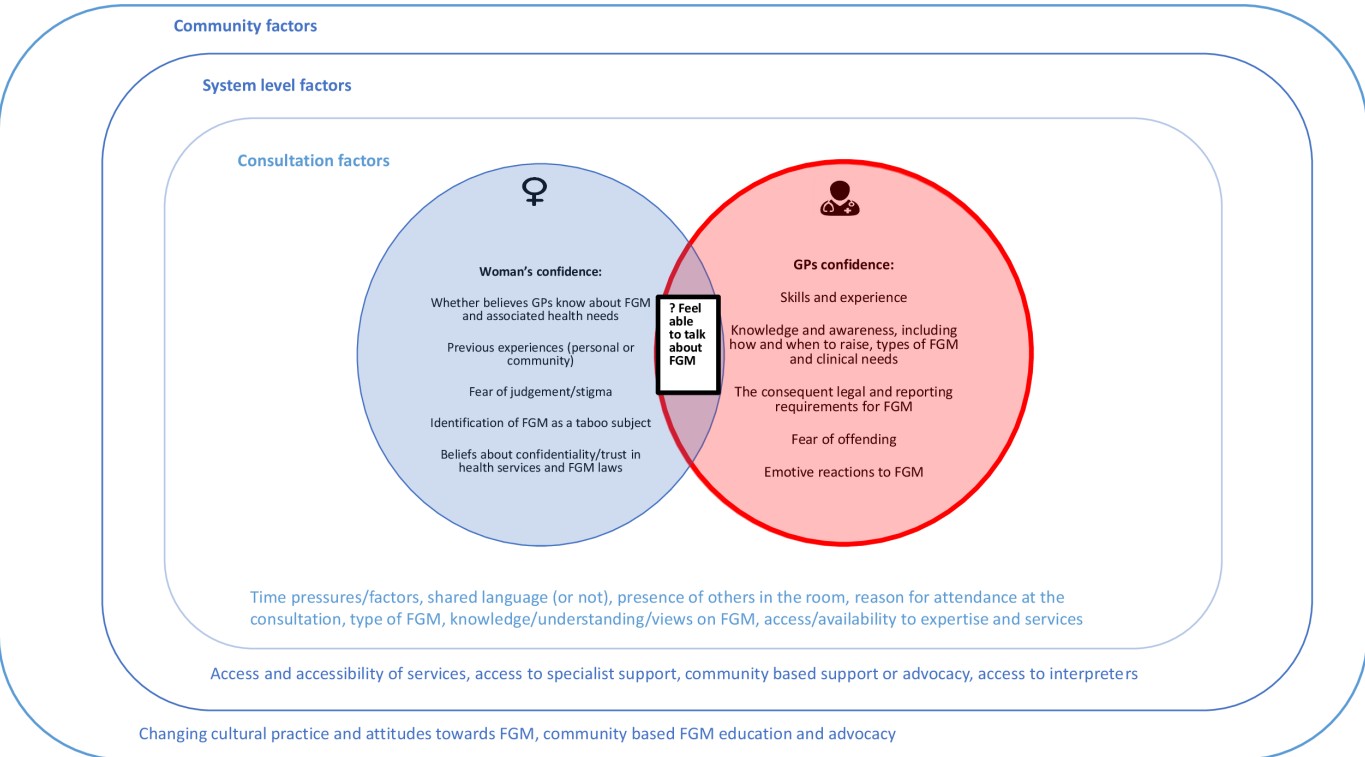

**Figure 2** Conceptual depiction of factors that may influence when GPs may or may not talk about FGM with their patients. FGM, female genital mutilation; GP, general practitioner.

Contexts:

Mechanisms:

Outcomes:

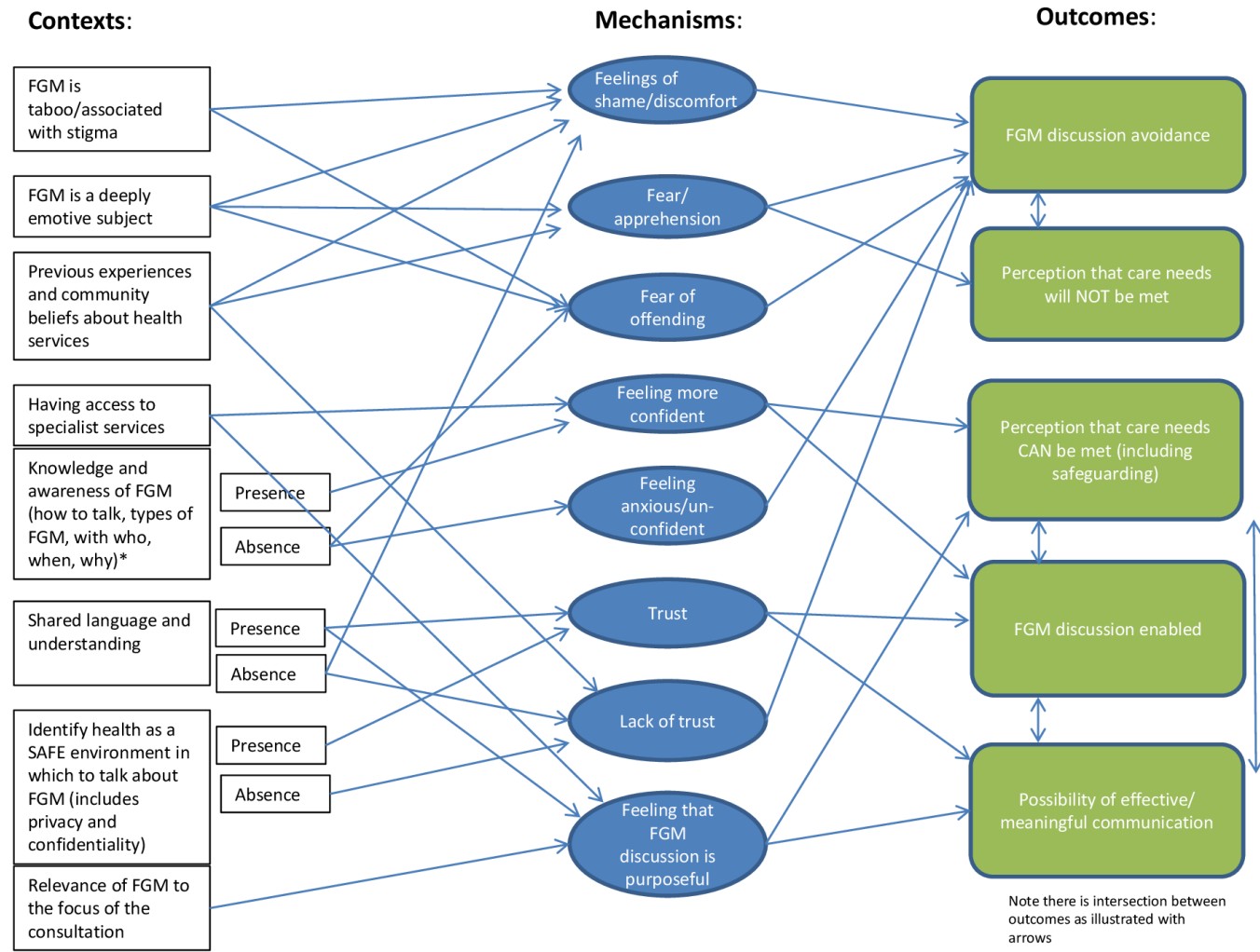

* This is a Continuum including awareness of lack of knowledge, and lack of this awareness

**Figure 3** Conceptual diagram illustrating the overarching programme theory for this synthesis. FGM, female genital mutilation.

## DISCUSSION
### Summary
GPs need adequate knowledge to support their patients with FGM, including the different FGM types and their different clinical presentations, needs and cultural contexts. This includes needing to be aware of local legislative, statutory and safeguarding requirements. GPs need skills to discuss FGM sensitively and with appropriate terminology. Language barriers can complicate conversations about FGM. Access to official interpreters is recommended, but they may not always be available. Even when available, there are potential pitfalls which GPs should be aware of, including consideration of who else is present in the consultation. The requirements of mandatory reporting and the FGM Enhanced Dataset may bring additional complications into the primary care consultation. Community health advocates could have a role in facilitating access to services.

### Strengths and limitations
As illustrated by table 1, only limited evidence was directly relevant to primary care. GPs have a vital role in managing FGM, yet there is little evidence about their

attitudes, knowledge or behaviour towards managing FGM in primary care, and none in the context of the 2015 policy changes. This synthesis therefore uses evidence from provider experiences in other healthcare settings, predominantly specialist clinics, and obstetrics and gynaecology services. Some challenges are likely to be comparable between these settings and primary care, namely, needing adequate knowledge and managing challenges with language and communication. However, there are differences between primary and secondary care that may limit this extrapolation, for example, that in obstetrics, FGM will almost always be relevant to the woman's reason for attendance, which is not the case in general practice. To address this lack of direct evidence, we have also undertaken a primary qualitative study with GPs.[144] The lack of primary data about GPs necessitated complex searches and we may have inadvertently not identified important evidence.

We identified limited evidence on the experiences and needs of women from outside of Africa and with forms of FGM other than type III. Given the evidence that these types may not be those that GPs most commonly

encounter, this is a potentially significant limitation. Whether these findings can be extrapolated to inform healthcare for women with different FGM types or from other FGM practising countries (eg, Malaysia or Indonesia) is unclear.

The GP consultation represents a coming together of GP and patient, and a strength of this synthesis is that it explores whether GPs talk about FGM using experiential evidence from both perspectives. Realist methodology allowed us to generate explanatory programme theory relevant to GPs from evidence on managing FGM in other healthcare settings and within other healthcare contexts, despite the lack of primary care data. Realist methodology supports the inclusion of grey literature. This project has benefitted from information on English community and GP perspectives reported by FGM advocacy organisations, which may not have been included in a traditional systematic review. Using iterative searching and citation tracking maximised data inclusion.

## Comparison with existing literature

The FGM literature is predominantly descriptive. This helps define potential challenges but offers less evidence about effective interventions with which to address them.

There is a rich literature about FGM, including a number of previous systematic reviews and qualitative metasyntheses, including a number cited within this synthesis as contributing evidence, perspectives and commentary relevant to this synthesis question and research objectives.[17 18 43–47 65 102] This review includes other syntheses, which included grey literature,[47] and evidence syntheses that included both provider and community perspectives.[45 46] In addition to their review of healthcare provider perspectives, Evans *et al*[145] have undertaken a comprehensive systematic qualitative synthesis of women affected by FGMs experiences of healthcare. This wide-ranging review included grey literature and considered experiences across all healthcare settings, including primary care. Their model of culturally safe care resonates with the realist theory we propose in this paper, as do their findings including the importance (and potential challenges) of effective communication, the value of positive encounters characterised by knowledge and compassion, and the adverse impacts on women when HCPs react to their FGM in ways which made them feel ashamed or judged.

This synthesis can add to this important developing body of literature by contributing a realist perspective, thus considering contextual and generative mechanisms across a range of processes potentially relevant to communication within the context of a primary care consultation. Using grey literature and opinion pieces and bringing together evidence from provider and community perspectives allowed us to triangulate possible influential factors. Finally, the realist methodology supported the inclusion of evidence from other research fields. This allowed us to postulate theory where there was no directly available evidence. For example, Evans *et al*[47] noted in their synthesis of factors relevant to healthcare provision of FGM healthcare from the perspectives of healthcare providers that there was no evidence about the delivery of safeguarding care. Using insights from evidence about IPV offered potential evidence to test our tentative CMOC with and support their development, for example. about what helps clinicians in responding to disclosures,[53 54] and about the tensions experienced by GPs in their role of coding into the longitudinal patient care record.[104–106] This synthesis adds consideration of the potential impacts of mandated reporting and data sharing on communication in primary care consultations.

In this synthesis, we present postulated programme theory about any factors that might be relevant in the dynamics of the primary care setting, which we hope will be further tested, appraised and improved.

Indeed, since we submitted this synthesis, a qualitative study exploring women with FGM's experiences of primary care in the Netherlands has been published, which is a welcome addition to the available literature. This study reported women's perceptions of challenges when seeing a GP, including concerns that their GP would not have adequate knowledge about FGM or would not be able to help them, and concerns about limited time in appointments. Satisfaction was increased when GPs were able to convey that they understood and were supportive of their care needs.[146] We consider that these findings resonate with the theory postulated in this synthesis. As documented previously, following and then in parallel with conducting this synthesis SD/SZ have conducted a qualitative study exploring GPs working in England's perspectives on supporting patients and families who might be affected by FGM. This study provides evidence which further supports the theory presented in this synthesis, including the need for holistic education about FGM, GPs concerns about identifying some forms of FGM on examination, and about knowing how to approach conversations about FGM with sensitivity and appropriate terminology. Holding responsibility for women's care needs in parallel with their responsibilities for managing any potential safeguarding needs in her family could be challenging for GPs. The mandatory reporting and FGM Enhanced Dataset reporting requirements could further complicate these care journeys, alongside concerns about the impacts of these duties on enduring primary care relationships. Specialist services were seen as critical for enabling these conversations in primary care.[144]

In their realist synthesis considering the experiences of UK maternity care by women with social risk factors, which included consideration of women with FGM, Rayment-Jones *et al*[147] also identified the importance of perceptions of kindness and respectfulness from HCPs, the value of trusting relationships and potential role for health advocates, and how factors such as language and access to interpreters contribute to the concept of candidacy for care. This review also identified fear of judgement by HCPs and perceptions of the health service role as surveillance rather than care were important

contextual factors relevant to accessing care. In their systematic review considering challenges and facilitators for refugees and asylum seekers in high-income countries, Robertshaw *et al*[148] also identified the importance of trusting relationships, acceptable and accessible interpreters where there were language barriers, and the importance of cultural competency in primary care, and the need for education and training to support this. A 2020 qualitative study exploring English healthcare providers' perspectives on the impacts of data sharing for immigration enforcement reported concerns about impacts on health access, patient/clinician relationships, and about the interface between policy and their professional ethics, notably confidentiality and trust.[149]

The lack of (and need for) an evaluation of mandatory reporting has been commented on by other authors.[150]

We have identified deficiencies in professional knowledge as an important contextual factor that can influence whether GPs talk about FGM. Other authors have noted the need for those in primary care to be informed to improve care for those affected by FGM.[139] There is little evidence to support interventions to improve how HCPs are enabled to support their patients with FGM.[151 152] One study with 11 midwives evaluated the effectiveness of an educational intervention and found that it was successful in promoting knowledge and confidence in managing FGM.[153] This intervention used learning from case studies which is suggested by community advocates to support FGM education.[14 55] A 2016 survey of medical students from five medical schools in London reported that the majority of respondents had not received formal teaching on FGM and were not aware of potential associated health issues. Having had formal education increased awareness, but despite this, only 50% of respondents who had been formally educated felt confident about identifying FGM on examination.[154] UK medical students have voted to ask for education about FGM. Medical students reported that after attending a workshop which included education on FGM, the UK law and how to talk about FGM, 75% of them felt more confident about communicating with a patient who had had FGM.[155] There is a call for FGM education to promote professionals' cultural competency.[18 29 33 156] Cultural competency education for HCPs offers likely benefits,[157] including for patients,[158] although formal cultural competency training is often lacking in general practice.[159] We have not identified any literature evaluating the impact of FGM education for primary care practitioners on their clinical confidence or cultural competencies. The need for research and evaluation on interventions to support caregivers in FGM has been noted previously.[152] A systematic review conducted in 2019 by three authors of this review (GH, SD and FG) found that, despite increases in FGM awareness in both healthcare and public spheres, HCPs remained subjectively and objectively undereducated and underprepared on the issue. While isolated countries such as Sweden had managed to target their education effectively, the majority (including high prevalence nations) struggled to approach FGM education and training adequately. Much of this stemmed from issues of cultural competency and confidence in knowledge, as is reflected in this study.[160]

That it can be important for health professionals to manage their own emotional reactions when they are supporting patients affected by FGM is resonant with research into IPV, which tells us that clinician responses, including blaming, judging or pitying, should be avoided.[95]

In tabulating the available evidence relevant to our synthesis question, we note that the available FGM evidence is predominantly from obstetric settings, with a lack of evidence from other settings, notably primary care, which is an important point documented in other systematic reviews.[17 47] The holistic life-course health needs for women with FGM, including their FGM-related needs outside of safeguarding, paediatric or obstetric settings, are important service and research needs.[16] We also note that much of the identified research considers the needs and experience of women with type III FGM, yet an English specialist paediatric clinic most commonly identified type IV FGM and no cases of infibulation and identified a girl from Malaysia with FGM.[75] That the existing evidence is potentially skewed towards type III FGM has been documented previously in a systematic review of healthcare providers experiences of caring for women with FGM.[47] It is important that clinicians are aware of the practice of FGM in some Asian countries (including Indonesia and Malaysia), although there is little evidence about prevalence to guide them.[75]

## Implications for research and practice

This review will help GPs (and GP educators) consider what knowledge or skills are needed to support GPs to feel confident to talk about FGM with their patients. It may help them consider the challenges when using interpreters to talk about FGM and highlights the potential challenges of managing the FGM reporting requirements.

Research is needed to explore what FGM-affected communities need from GPs and primary care. This needs to include all types of FGM, and all communities that practice FGM, and all aspects of access to and experiences of care, whether directly related to FGM or not.

There is a need to research and understand FGM-related health needs throughout the life course, including, for example, the needs of women living with FGM throughout and after their menopause. Expert commentators have noted the lack of evidence underpinning some of the guidance that clinicians are offered about what signs they should look for to try to prevent or anticipate FGM.[75] Clinicians urgently need evidence that will help them protect girls and families, including consideration about what policies and strategies are acceptable and effective within communities. We need evidence to inform this and believe that this needs to be developed in partnership with community members and front-line clinicians.

The effectiveness of policies and legislation on deterring community practice and working towards eliminating the practice of FGM are vital considerations but which were beyond the scope of this review, which focused on their impacts on communication in consultation. However, the potential impacts of the mandatory reporting and FGM Enhanced Dataset requirements on healthcare interactions do need to be evaluated, including HCP and community perspectives.

Educational interventions, research and services should be developed in partnership with community members, using their expertise and experience, to ensure resources meet their needs.

**Contributors** SD and FG wrote and developed the study protocol and the initial programme theory, and conducted the data analysis and synthesis. SZ contributed to the study design and development, writing and data presentation advice and expertise, analysis and theory development. CP provided methodological guidance and advice. CD and GH read and agreed on abstracts and data for final inclusion. CD assisted with study searching and data management. All authors contributed to the development and writing of this article.

**Funding** SD was funded by a National Institute for Health Research (NIHR) in-practice fellowship (IPF-16-10-03, 2016-2018) during this research project. CD is supported by an NIHR Research Methods Programme Systematic Review Fellowship (NIHR-RM-SR-2017-08-018). SZ is an NIHR Senior Investigator. The views expressed are those of the author(s) and not necessarily those of the NHS, the NIHR or the Department of Health and Social Care.

**Competing interests** SD is a trustee of Oxford Against Cutting. Following this project, she has acted as the Royal College of General Practitioners college representative for female genital mutilation. SD has held small grants to develop and report on the patient and public involvement project that underpinned this research. There are no other author competing interests to declare.

**Patient consent for publication** Not required.

**Provenance and peer review** Not commissioned; externally peer reviewed.

**Data availability statement** All data relevant to the study are included in the article or uploaded as supplemental information. For this realist review we analysed published literature that is in the public domain.

**ORCID iDs**
Sharon Dixon http://orcid.org/0000-0002-7469-6093
Claire Duddy http://orcid.org/0000-0002-7083-6589
Frances Griffiths http://orcid.org/0000-0002-4173-1438

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
