## [Reviewer comments · BMJ Open]

ARTICLE DETAILS

TITLE (PROVISIONAL)	Conversations about FGM in primary care: A realist review on how, why, and under what circumstances FGM is discussed in general practice consultations.
AUTHORS	Dixon, Sharon; Duddy, Claire; Harrison, Gabrielle; Papoutsis, Chrysanthi; Ziebland, Sue; Griffiths, Frances

VERSION 1 – REVIEW

REVIEWER	Andreas Balthasar Department of Health Science and Medicine University of Lucerne Switzerland
REVIEW RETURNED	03-Jun-2020

GENERAL COMMENTS	Very valuable paper based on a comprehensive literature analysis and a very suitable method (realist synthesis). The title is not appropriate. It should be chosen more meaningful, since the developed program theory includes both the side, the affected persons and the GP. Furthermore, the program theory should not only be sketched but also described. In particular I miss a weighting of the relevance of the different CMOC.
--

REVIEWER	Dr Georgina Davis Canterbury Hospital Sydney Local Health District Canterbury Road, Campsie Sydney NSW Australia
REVIEW RETURNED	05-Jun-2020

GENERAL COMMENTS	Thank you for the opportunity to review this manuscript. An extraordinary amount of work has gone into collating, reviewing and synthesising the existing literature on an important topic in Women's Health and the manuscript should be accepted for publication A few minor comments - The title only states "When do GPs talk about FGM?" The manuscript covers so much more than the "when" so perhaps a more inclusive title addressing the circumstances of the when how why as outlined in the definition of a realist synthesis should be considered.- Page 5 Line 51 & Page 6 Line 31. Creighton SM "FGM - What every paediatrician should know" Arch Dis Child 2016 may be a nice reference to consider in addition to the information from the Obstetrics and Gynae community.
---

	- The Figures weren't labelled - I'm not sure if this was a formatting issue on how the proof was presented - Page 41 Line 8 - a highlighted note/comment from authors to each other remains.
--	---

REVIEWER	Catrin Evans University of Nottingham
REVIEW RETURNED	08-Jun-2020

GENERAL COMMENTS	Thank you for the opportunity to review this paper. It is an extremely comprehensive and thorough piece of work and has been well executed methodologically. I just have a few comments: The title does not seem to quite match up with the way that the results are presented. i.e. the title asks “when do GPs talk about FGM with their patients?”.... But the results do not really answer this questions (and certainly not in the abstract). Rather, the review presents various explanations as to why GPs may, or may not, discuss the issue of FGM (rather than focusing on ‘when’). I would suggest amending the title. The paper frames the entire issue around primary care and GPs in England. Given the global audience of the journal and also the international literature that was used to inform the synthesis, it would be good to see the introduction and discussion also consider the relevance of the issue and of the synthesis to primary care contexts internationally. In addition, given that primary care in the UK is changing rapidly to include other health professionals (e.g. advanced nurse practitioners) who may have first contact with patients, this may need to be acknowledged. In terms of methods, in the section on searching, I would expect to see some commentary about existing systematic reviews around this topic (perhaps focusing on what is known about healthcare professionals rather than just GPs) and an analysis of what can already be inferred about the topic from this very considerable pre-existing body of evidence. I would then expect to see the Discussion reflect upon what the realist synthesis has added that was not already known. I appreciate the huge amount of work that has gone into this synthesis, however, given the large existing body of evidence and, particularly, reviews, on this topic, I would like to see a much clearer and stronger articulation of how this review using this methodology has added new insights – and then more discussion on how these new insights might inform policy and practice. Also in the methods section, it is noted that there was a level of PPI involvement. Can the authors elaborate on the nature of the PPI involvement (i.e. who they were and how they contributed) and what difference the PPI involvement made? – i.e. more of a reflexive consideration of the whole PPI involvement? In line with this reflexive approach, I think it is also important to include more information on the research team and why this review was undertaken. I felt that the implications section was a little short and rather weak, given the level of detail presented in the synthesis. It would be particularly useful to see some more consideration in the implications around the mandatory recording policy given that data is consistently showing that FGM does not appear to be taking
--

	place in the UK (but rather has usually been performed prior to arrival in the UK).
--	---

VERSION 1 – AUTHOR RESPONSE

Reviewer 1

Very valuable paper based on a comprehensive literature analysis and a very suitable method (realist synthesis). The title is not appropriate.

We have proposed an amended title.

It should be chosen more meaningful, since the developed program theory includes both the side, the affected persons and the GP. Furthermore, the program theory should not only be sketched but also described. In particular I miss a weighting of the relevance of the different CMOC:

Thank you for this advisory comment. We agree that these final summary figures postulating the potential relationships between our emergent CMOCs are better framed with an adequate introduction and heading. We hope that the one we have drafted within the article text is acceptable.

The question about weighting is a very interesting one, and one which we had considered carefully then, and are glad to have the opportunity to re-visit. We recognise that weighting is an area of debate and reflection in realist methodology. This review question, with its subsequent derived programme theory, focussed on the coming together of those seeking care for FGM in primary care with those who could provide this care. We envisaged this process as a journey, or sequence of tasks, all of which potentially need to be successfully met or navigated, to achieve an ultimate outcome of achieving effective communication (as a pathway to accessing effective and accessible care). Navigating this journey could succeed or fail at any step, and therefore we felt that weighting some factors as more important than others could lose this sense of value for consideration of all potentially relevant factors. While there were some CMOC for which there was more evidence, we consider that one of the unique strengths of the realist approach used here is in creating a forum where we can align and reflect on a wide range of possible factors.

Reviewer 2:

- The title only states "When do GPs talk about FGM?" The manuscript covers so much more than the "when" so perhaps a more inclusive title addressing the circumstances of the when how why as outlined in the definition of a realist synthesis should be considered.

Thank you for this feedback. Please see above for a proposed title amendment. .

- Page 5 Line 51 & Page 6 Line 31. Creighton SM "FGM - What every paediatrician should know" Arch Dis Child 2016 may be a nice reference to consider in addition to the information from the Obstetrics and Gynae community.

Thank you so much for the guidance about going back to this paper. This was initially identified and reviewed as an abstract. The guidance about examination was identified as important but was already represented in our review with the original data/empirical data from the case series papers published by the same authors and was therefore not included. The reflections on examining in the context of a child protection medical were not identified as relevant to an English GP, who would refer to an expert paediatrician for such an examination. However, on reviewing the paper at your suggestion, we consider that the reflections on education to support mandatory reporting are relevant and have

therefore included this. This also prompted a further review of an additional paper by the same authors, designated for all health professionals, which explicitly highlights the importance of awareness for primary care, and the potential impacts of legislation on communication. In keeping with realist methodology, allowing iterative re-appraisal of evidence, I am grateful to you for this suggestion, and have incorporated this into the paper.

With Many Thanks.

- The Figures weren't labelled - I'm not sure if this was a formatting issue on how the proof was presented

We hope that this was a formatting issue and will seek to address this. In response to another reviewer, we have additionally drafted an introductory paragraph to introduce the figures, which we hope will also add clarity.

- Page 41 Line 8 - a highlighted note/comment from authors to each other remains. Our sincere apologies. This will of course be resolved.

Reviewer 3:

The title does not seem to quite match up with the way that the results are presented. i.e. the title asks "when do GPs talk about FGM with their patients?".... But the results do not really answer this questions (and certainly not in the abstract). Rather, the review presents various explanations as to why GPs may, or may not, discuss the issue of FGM (rather than focusing on 'when'). I would suggest amending the title.

Thank you for this feedback. Please see above for a proposed title amendment.

The paper frames the entire issue around primary care and GPs in England. Given the global audience of the journal and also the international literature that was used to inform the synthesis, it would be good to see the introduction and discussion also consider the relevance of the issue and of the synthesis to primary care contexts internationally.

In addition, given that primary care in the UK is changing rapidly to include other health professionals (e.g. advanced nurse practitioners) who may have first contact with patients, this may need to be acknowledged.

Thank you. We agree that it is a useful addition to this paper to include an additional explanation about the English primary care context to facilitate readers working in other contexts to appraise the potential relevance of this review to their own area of practice. We have added a description of some of the core aspects of primary care in England as relevant to this review. We note that this can be different in different health care settings and different countries. We hope that defining this, as the core concepts of primary care that under-pinned our approach to data analysis, review and synthesis to programme theory development, will allow international readers to reflect on which areas are relevant to their own practice or healthcare setting. We agree that the setting of primary care, with its longitudinal care (and care records), generalist long-term care, and gatekeeper function, embedded in communities is the core of what we were working with in this synthesis, and that this is therefore potentially applicable across all health professionals working in primary care. We recognise that clarifying who usually delivers primary care is important, to make clear the context in which we approached this synthesis. We note that this has been the model used in a primary care paper cited in this study (Kaplan-Marcusan A, Torán-Monserrat P, Moreno-Navarro J, Fàbregas MJC, Muñoz-Ortiz L. Perception of primary health professionals about Female Genital Mutilation: from healthcare to intercultural competence. BMC Health Services Research. 2009;9(11).), in which their definitions of

the structure of primary care in Spain helped us to consider how these related to our synthesis. We hope that adding a clarifying sentence to this effect in the introduction is an acceptable means of addressing this point.

Furthermore, it is important to note that we explicitly considered the ways in which primary care might differ from the settings in which empirical evidence we considered was derived; as we comment in our paper, FGM is usually immediately potentially relevant to the woman's current/presenting care needs in obstetrics and gynaecology or at a specialist FGM clinic. This is not the case in primary care, and so we postulate that expectations of what will/should/could be discussed may differ. This synthesis approach sought to consider this, and so the context of English primary care was central to our synthesis. For obstetricians and midwives (and usually paediatricians also) there is usually the context of a clinical relationship which is predominantly with the woman (or index patient) who presents, and which endures for the duration of that clinical need. This is not the case with primary care, GPs hold enduring relationships, and the GP might have clinical relationships and direct responsibilities for other members of their families if they are registered at the same practice. This includes but potentially extends beyond safeguarding.

In terms of methods, in the section on searching, I would expect to see some commentary about existing systematic reviews around this topic (perhaps focusing on what is known about healthcare professionals rather than just GPs) and an analysis of what can already be inferred about the topic from this very considerable pre-existing body of evidence. I would then expect to see the Discussion reflect upon what the realist synthesis has added that was not already known. I appreciate the huge amount of work that has gone into this synthesis, however, given the large existing body of evidence and, particularly, reviews, on this topic, I would like to see a much clearer and stronger articulation of how this review using this methodology has added new insights – and then more discussion on how these new insights might inform policy and practice.

Thank you. This is a very valid and useful commentary. We have made changes in the text in response to this and would of course welcome further expert feedback on this.

Also in the methods section, it is noted that there was a level of PPI involvement. Can the authors elaborate on the nature of the PPI involvement (i.e. who they were and how they contributed) and what difference the PPI involvement made? – i.e. more of a reflexive consideration of the whole PPI involvement? In line with this reflexive approach, I think it is also important to include more information on the research team and why this review was undertaken.

The PPI project that SD developed and (with PPI collaborators) reported on under-pinned this synthesis, contributing significantly to the development of the research question and project aims, and to the development of the initial programme theory. The evolving programme theory was reviewed with expert stakeholders, with expertise in FGM, primary care, or both. In particular, PPI input focussed the review towards communication in primary care, the skills needed by GPs, and that the experiences and perspectives of community members accessing primary care needed to be considered and included. The importance of also considering this within the context of the policies of mandatory reporting and the FGM Enhanced Dataset arose from this PPI.

The findings of this synthesis were presented by SD and then discussed at a local FGM strategy group meeting in January 2019, at which there were health professionals across a range of specialisms (midwifery, obstetrics, paediatrics, public health, health visiting), social workers, community member representatives, and local advocacy organisation representation. These points have been clarified in the text.

I felt that the implications section was a little short and rather weak, given the level of detail presented in the synthesis. It would be particularly useful to see some more consideration in the implications

around the mandatory recording policy given that data is consistently showing that FGM does not appear to be taking place in the UK (but rather has usually been performed prior to arrival in the UK).

Thank you. We think that this is an important implication and also an emergent research priority. We have expanded this section, in response to this helpful feedback.

VERSION 2 – REVIEW

REVIEWER	Andreas Balthasar University of Lucerne, Health Science & Medecine, Switzerland
REVIEW RETURNED	23-Oct-2020

GENERAL COMMENTS	My remarks are included in the revised version. Thanks
--

REVIEWER	Catrin Evans University of Nottingham
REVIEW RETURNED	10-Oct-2020

GENERAL COMMENTS	Thank you for an excellent paper and contribution to the field.
---